# The Transport Properties of Semi-Crystalline Polyetherimide BPDA-P3 in Amorphous and Ordered States: Computer Simulations

**DOI:** 10.3390/membranes12090856

**Published:** 2022-08-31

**Authors:** Alexey Y. Dobrovskiy, Victor M. Nazarychev, Igor V. Volgin, Sergey V. Lyulin

**Affiliations:** Institute of Macromolecular Compounds, Russian Academy of Sciences, Bolshoj pr. 31 (V.O.), 199004 St. Petersburg, Russia; aleksey_dobrovsky@mail.ru (A.Y.D.); nazarychev@imc.macro.ru (V.M.N.); i.v.volgin@gmail.com (I.V.V.)

**Keywords:** polyimide, gas separation, membranes, structure ordering, molecular dynamic, simulations

## Abstract

The effect of polymer chain ordering on the transport properties of the polymer membrane was examined for the semi-crystalline heterocyclic polyetherimide (PEI) BPDA-P3 based on 3,3′,4,4′-biphenyltetracarboxylic dianhydride (BPDA) and diamine 1,4-bis [4-(4-aminophenoxy)phenoxy]benzene (P3). All-atom Molecular Dynamics (MD) simulations were used to investigate the gas diffusion process carried out through the pores of a free volume several nanometers in size. The long-term (~30 μs) MD simulations of BPDA-P3 were performed at T = 600 K, close to the experimental value of the melting temperature (Tm ≈ 577 K). It was found during the simulations that the transition of the PEI from an amorphous state to an ordered one occurred. We determined a decrease in solubility for both gases examined (CO_2_ and CH_4_), caused by the redistribution of free volume elements occurring during the structural ordering of the polymer chains in the glassy state (Tg ≈ 487 K). By analyzing the diffusion coefficients in the ordered state, the presence of gas diffusion anisotropy was found. However, the averaged values of the diffusion coefficients did not differ from each other in the amorphous and ordered states. Thus, permeability in the observed system is primarily determined by gas solubility, rather than by gas diffusion.

## 1. Introduction

Membrane gas separation methods have received a lot of attention in different areas of industry over the past few decades. Membrane technologies are becoming more science-intensive each year, making it possible to solve complex problems, and in particular, the purification of natural gas from various impurities. The development of new materials for the selective transport of gases is an increasingly urgent task, because of the constant increase in industrial demand for new membrane materials [1,2]. Currently, gas separation is attracting the attention of various industrial companies [3,4,5,6,7] which make use of different polymers as gas separation membranes [8,9,10,11].

The materials used for gas separation should be both highly permeable and highly selective. However, it is well known that more permeable membranes are usually less selective, and vice versa [12,13]. In this regard, the development of materials to overcome this permeability/selectivity trade-off becomes an outstanding challenge. Among the various polymer classes, polyetherimides (PEI)s are the most promising as membrane gas separation materials, due to their stability in the face of extreme temperature changes and aggressive environments, outstanding film-forming and mechanical properties, their resistance to ageing, and their thermal and chemical stability [8,9,10,14,15,16,17,18]. In terms of their transport properties, a large number of PEIs do not exhibit particularly high permeability values, although they do display good selectivity for various gas pairs [19,20], in particular CH_4_/CO_2_, which makes their use attractive for gas separation.

The repeating unit of a PEI consists of two fragments: diamine and dianhydride. The extensive possibility of varying the chemical structure of these fragments allows researchers to develop materials with controlled physical properties. PEIs can be divided into two types: completely amorphous, and semi-crystalline, i.e., containing both amorphous and crystalline domains. The transport of gases in semi-crystalline polymers is usually described by a two-phase model, whereby the diffusion of gas molecules occurs through the amorphous phase in the polymer, while the crystalline domains remain practically impermeable to the penetrant [9,16]. The diffusion coefficient in the crystalline region is orders of magnitude smaller than the equivalent value in the amorphous region. As such, the presence of ordered domains can have a pronounced effect on the membrane’s permeability and the gas separation process itself. The assumption of the applicability of the two-phase model has been widely confirmed by experimental data [21,22]. It has been shown that, at a low degree of crystallinity, its influence on diffusion is small [21], while as the crystallinity increases, the diffusion of gases usually decreases [23]. The use of membranes with a low degree of crystallinity makes it possible to maintain a balance between the gas permeability of the membrane and its mechanical properties. Nevertheless, in some cases the physical model of diffusion in semi-crystalline polymers appears to be oversimplified, since the sorption behavior of the amorphous phase can be affected by the constraints imposed by the presence of crystalline domains distributed in the sample, as well as by the behavior of the interphases [22,24]. Indeed, it has been shown that for semi-crystalline syndiotactic polystyrene, the contribution of the crystalline phase to the total carbon dioxide solubility is 20 times higher than that of the amorphous phase [24]. Due to the fact that the solubility of gases directly affects their permeability, it can be concluded that the presence of ordered domains may have a positive effect on the permeability of the membrane. Moreover, some studies claim that gas transport can occur in polymers with a high degree of crystallinity [25,26].

It should be noted that the times required for observing the crystallization of polymers in MD computer simulations turn out to be practically impossible for researchers. In this regard, amorphous and ordered states are most often considered in simulations. The latter precedes the process of polymer crystallization and could lead to an improvement in mechanical properties, while still maintaining transport properties. The effect of ordering on mechanical properties has been studied both experimentally [27] and in computer simulations in our previous studies [28,29].

As noted earlier, the chemical structure of the repeating unit has a strong influence on the physical properties of the polymer. For example, the introduction of hinged atoms into the polymer backbone could lead to the appearance of thermoplasticity in PEIs and decrease their glass transition temperature (*T_g_)* [30]. The synthesis of thermoplastic semi-crystalline PEIs with low viscosity in melt has been carried out at the Institute of Macromolecular Compounds of the Russian Academy of Sciences, where semi-crystallizable polyetherimide R-BAPB based on 1,3-bis(3,3′,4,4′-dicarboxyphenoxy)benzene (dianhydride R) and 4,4′-bis(4-aminophenoxy)biphenyl (diamine BAPB) were first synthesized. R-BAPB has been extensively studied both experimentally and in computer simulations [31,32,33,34,35,36].

It was proven that even a slight variation in the chemical structure could have a profound effect on the crystallinity of a polymer [28,30,37,38]. In our previous studies, we investigated the mechanical and thermophysical properties of PEIs developed by the group of Prof. Dingemans and the nanocomposites based upon them, and obtained qualitative agreement between the simulation and the experimental data [28,29,39,40]. Investigation of the structural ordering of polymer chains, leading to subsequent crystallization of the polymer using computer simulations, turns out to be a CPU-intensive task. Even for molecules with simple chemical structures such as the alkanes C_20_H_42_, C_50_H_102_, and C_100_H_202_, it was shown that crystal growth rate is approximately ~10^−4^–10^−1^ nm/ns, and therefore computer simulation of crystallization is very resource-hungry [41].

PEIs have a much more complex chemical structure than alkanes, which includes various aromatic fragments and heterocycles. Due to this complexity, crystallization of PEIs occurs at much longer time values than those of alkanes, and such times turn out to be inaccessible in simulations. Therefore, Molecular Dynamics (MD) simulations allow us to investigate the amorphous and ordered (pre-crystallization) stages of PEIs. We have previously studied the structural ordering of polymer chains of R-BAPB [39] and BPDA-P3 [28] polyetherimides near the surface of a carbon nanotube. The repeating unit of the crystallizable thermoplastic BPDA-P3 is based on 3,3′,4,4′-biphenyltetracarboxylic dianhydride (BPDA) and diamine 1,4-bis[4-(4-aminophenoxy)phenoxy]benzene (P3) (see Figure 1). For BPDA-P3 filled with nanotube, the initial stage of crystallization was established: their polymer chains were ordered relative to each other [28]. Such ordering led to a significant improvement in the mechanical properties of both polymers. BPDA-P3 chains can be ordered relative to each other without a carbon nanofiller, and it also has a high glass transition temperature (*T_g_* ≈ 487 K). Moreover, PEIs with BPDA dianhydride have good potential for gas separation [42].

The study of the transport properties in both amorphous and ordered states is an important task, since the ordering of polymer chains determines the polymers’ supramolecular structure which, in turn, strongly affects the physical properties of the polymers themselves. It has been shown that the orientation and ordering of the PEI chains is one of the most effective ways to increase the selectivity of PEI-based membranes (the property of the membrane having different permeabilities for the different components of the mixture to be separated) [43,44]. Such an effect may be associated with the redistribution of the free volume, which leads to the remarkable gas separation characteristics of the PEI-based membranes. However, questions relating to the ordering dependence of the diffusion, solubility and permeability remain open: to what extent do they differ in their amorphous and ordered states. It is a generally accepted point of view that during the ordering the permeability should decrease, however it is necessary to establish which parameter is crucial for the change in permeability: the solubility or the diffusion coefficient. Investigation of the reason for the difference in BPDA-P3 transport properties in amorphous and ordered states is the purpose of this study.

## 2. Materials and Methods

### 2.1. Simulation of BPDA-P3 Structural Ordering

Microsecond-scale MD computer simulations using all-atom models are a state-of-the-art approach [45] enabling the study of the different physical properties of polymers. Moreover, such a method is useful for studying the process of the structural ordering of polymer chains at temporal microscales and dimensional nanoscales. We have previously shown that the introduction of a nanofiller (single-walled carbon nanotubes) into a polymer matrix could lead to the reduction of the time required for the structural ordering of BPDA-P3 chains relative to each other near the surface of the carbon nanotube [28]. In the case of nanocomposites, the carbon nanofiller acts as a crystallization initiator (nucleation agent), and accelerates the onset of ordering. However, in an unfilled sample ordering should occur on its own, which leads to an increase in the simulation time required for the transition of the polymer from an amorphous state to an ordered one—up to several tens of μs!

The use of coarse-grained models can significantly reduce the time required for computer simulations compared to all-atom representation. This approach is often used in the study of various polymers [46,47,48,49,50,51,52,53], in particular polyetherimides (PEI), and nanocomposites based on them. However, a reverse mapping procedure may provide incorrect free volume distribution on the atomic scale and in any case, long, atomistically detailed simulations are necessary to validate such models.

The generation of the initial configurations of BPDA-P3 in the simulation box was carried out using the approach developed previously [36,39,40,45,54]: 27 polymer chains of BPDA-P3 with a degree of polymerization *N_p_* = 8 were randomly placed into a periodic box. Compression of the system initially created and three repeating cycles of annealing from 800 K to 300 K of the system were performed. After annealing, the system was cooled to 600 K with the following equilibration run (~1.5 μs) at *T* = 600 K. We have previously shown that during 1.5 μs of equilibration at *T* = 600 K the average BPDA-P3 chain sizes reach equilibrium time-independent values which are in agreement with the theoretical estimations [40].

Long-term (~30 μs) computer simulations at *T =* 600 K were performed using the GROMACS 5.0.7 MD package [55,56]. This temperature is slightly higher than the experimental melting temperature of BPDA-P3 *T_m_* ≈ 577 K [27]: at this temperature, the mobility of the polymer chains allowed us to obtain a state in which the BPDA-P3 chains can orient themselves relative to each other [28]. We should mention that the simulation value of the melting temperature in case of all-atom force fields was slightly higher than the experimental one, possibly due to the difference in the heating rate [57]. In order to maintain a constant temperature and pressure (*NpT* ensemble), a Berendsen thermostat and barostat were used with the time constants *τ_T_* = 0.1 ps and *τ_p_* = 0.5 ps, respectively, similar to our previous simulations [28,45]. The simulation step was selected to be equal to 1 fs. Periodic boundary conditions were used in all three dimensions of the simulation box with initial sizes 5.9 × 5.9 × 5.9 nm.

The set of parameters for potential functions corresponded to the Gromos53a6 force field, which has been used successfully to simulate the thermal and mechanical properties of thermoplastic PEIs [58,59], and in particular BPDA-P3 [28,40]. This force field was able to reproduce the structural ordering (pre-crystallization) process of polymer chains in nanocomposites [28,38,39], and to provide a coincidence with the experimental data [27,37]. The partial charges were equal to zero for two reasons: firstly, a repeating unit of BPDA-P3 does not have strong polar groups (where the significant electron density shift occurs) between which additional considerable dipole-dipole interactions can arise; secondly, we have shown previously that taking into account the partial charges in the simulation of such PEIs did not significantly affect the general character of the structural ordering of the planar PEI fragments [38]. Moreover, as was shown in our earlier studies, qualitative agreement of simulation data with experimental data on structural and mechanical properties can also be observed when using models that do not take into account the presence of partial charges [39,40]. Thus, in the absence of strongly polar groups in the polymer, the contribution of partial charges may not be very important. Moreover, our previous investigation of the PEI transport properties without taking into account partial charges provide a quantitative agreement between the transport properties obtained in simulations and in the experiment [34]. In this regard, the corresponding partial charges were set to zero, which made it possible to significantly reduce the computational resources required for the study. The influence of partial charges on transport properties is a separate and important task which is beyond the scope of the current study. Nevertheless, the time required to simulate such a long trajectory (~30 μs) with zero partial charges was, according to our estimation, more than ~ 5 × 10^5^ processor-hours, or approximately one and a half years of continuous simulations on Xeon E5 v3 processors utilizing 64 CPU cores. Such long all-atom simulations were necessary for the onset of ordering of BPDA-P3 chains as a whole.

### 2.2. BPDA-P3 Transport Properties

To study the transport properties of BPDA-P3, we chose 6 independent configurations: 3 in the amorphous state and 3 in the ordered state from a 31 µs-long trajectory. 6 systems conforming to these configurations were cooled down from 600 K to 300 K with an effective cooling rate *γ_c_* = 1.5 × 10^11^ K/min at pressure *P* = 1 bar [28,40,60]. During the cooling process, for each of the 6 systems we obtained 19 different configurations at temperatures below *T_g_*: from 480 K down to 300 K with temperature steps equal to 10 K (*T_g_* ≈ 487 K [27]). These configurations were used for further calculation of the gases’ solubility and diffusion coefficients.

It is worth noting that in our previous studies dedicated to computer simulations we showed that all-atom force fields could over- or underestimate the values of different physical characteristics, due to the enormous difference in the cooling rates employed in the simulations and experiments [54,57,61,62]. Using previous MD simulations results we showed that the glass transition temperature of polyetherimide R-BAPS based on dianhydride R and diamine 4,4′-bis(4″-aminophenoxy)biphenyl sulphone) (BAPS) increases with increasing cooling rate *γ_c_* [54]. Furthermore, the cooling rate could influence the mechanical properties of the polymers: the results obtained demonstrate that for PEI R-BAPO based on dianhydride R and diamine 4,4′-bis-(4″-aminephenoxy)diphenyloxide (BAPO), the yield stress *σ_y_* decreases with the increase of *γ_c_* [59]_._

Neyertz et al. developed special methodology to estimate the number of penetrant molecules required for insertion into polymer samples to calculate diffusion properties [63]. In order to avoid polymer swelling, only 10 CO_2_ or CH_4_ gas molecules were added to each system to calculate gas diffusion coefficients in BPDA-P3, similar to our previous study [34]. Gas molecules were randomly inserted into unoccupied cavities of free volume in the simulation box. The insertion of gas molecules into the polymer matrix was performed using the GROMACS routine *gmx insert-molecules* [55]. Parameters *σ* and *ε* for the gas molecules were taken from the study of the transport properties of R-BAPB, where we used the additional parameters of special atomic types for the all-atom models of gas molecules (He, N_2_, O_2_, CH_4_ and CO_2_), due to their absence in the Gromos53a6 force field [34]. Parameters for the CH_4_ molecule were calibrated for thermal, structural, and dynamic properties by Yin and MacKerell [64]. Standard combination rules were used to calculate the non-bonded parameters for “gas-polymer” interactions [65,66]. It was shown that the implementation of such parameters for gas molecules into Gromos53a6 allowed us to reproduce quantitative trends and correlations between the solubility and diffusivity of the gases observed in the experiment and simulations for R-BAPB [34].

After the gas molecules were inserted, the steepest descent algorithm was used to minimize the potential energy of the resulting systems consisting of PEI and molecules of gases. A subsequent short 1 ns-long simulation was performed in the NVT ensemble to ensure that the energy of the systems was properly relaxed. We have previously shown that 1 ns is sufficient time for energy relaxation in such systems [34]. Long-term simulations of 500 ns duration were performed for each system to calculate the gas diffusion coefficients. Since the Berendsen thermostat suppresses fluctuations in kinetic energy exponentially, it can lead to an incorrect distribution of the kinetic energy and, accordingly, an incorrect distribution of velocities in the system [56]. This can be critical for calculating the diffusion properties of the system, and therefore we replaced it with the Nosé–Hoover thermostat [67,68] with the time constant *τ_T_* = 0.5 ps. For pressure coupling, the Berendsen barostat was also replaced with the Parrinello–Rahman [69] barostat with the time constant *τ_p_* = 2.5 ps. These parameters allowed us to investigate the physical properties of PEIs in previous studies [28,40,70].

### 2.3. Transport and Structural Properties Calculations

In MD simulations [63,71], the solubility *S* may be estimated using several widely used approaches, such as Grand Canonical Monte Carlo (GCMC) method [72,73,74], free energy method [75], and Widom (Test Particle Insertion) method [76], each of which is used for specialized tasks. In the Test Particle Insertion (TPI) approach a test molecule is inserted randomly into a polymer configuration, and the excess chemical potential *µ_ex_* could be calculated as [63]:
(1)
$$\mu_{ex}=-k_{B}T\cdot ln\langle exp\left(-\Delta U/k_{B}T\right)\rangle ,$$

where *ΔU* is the change in the total potential energy, corresponding to the insertion of the one test particle, 〈 〉 indicate averaging over the number of test insertions. 

Widom’s Test Particle Insertion scheme successfully used to determine the solubility of small molecules in different systems in agreement with experimental results [34,72,77]. GROMACS software with *tpi* integrator [56] was used to perform TPI calculations [78].

The solubility *S* can be expressed as [9]:
(2)
$$S=\frac{V_{mol}}{RT}\cdot exp\left(\frac{-\mu_{ex}}{RT}\right),$$

where *V_mol_* is the molar volume of the gas at standard temperature and pressure conditions (STP, *T* = 273.15 K and *P* = 1 bar), and *R* is the gas constant.

The diffusion coefficient *D* of the gas molecules could then be calculated using the Einstein equation:
(3)
$$\Delta r_{com}^{2}\left(\Delta t\right)=6D\Delta t,$$

where 
$\Delta r_{com}^{2}\left(\Delta t\right)$
 is the mean-squared displacement (MSD) of the center of mass of the gas molecules.

In Equation (3) it is assumed that gas molecules move chaotically in three-dimensional space in the normal diffusion regime. This regime could be observed after both the ballistic regime (when the MSD is proportional to *t^2^*) and the subdiffusive or anomalous regime (when MSD ~ *t^n^*, 0 < *n* < 1 [9,63,71,79]). Determination of the diffusion coefficient at low temperatures is even more complicated, since the mobility of the polymer chains decreases with decreasing temperature, which leads to an increase in the simulation time required to reach the normal diffusion regime. According to the activation approach to describe gas diffusion in polymers, the temperature dependence of the diffusion coefficient corresponds to the semi-empirical Arrhenius law [80]:
(4)
$$D=D_{0}\cdot exp\left(\frac{-E_{D}}{RT}\right),$$

where *D_0_* is the pre-exponential factor, and *E_D_* is the diffusion activation energy.

The permeability *P* of a polymer film for a certain gas can be expressed as [9]:
(5)
P=S·D.


The distribution of free volume in the system was determined using the Hofmann et al. “V_connect” method [81]. The simulation box was divided into a grid with step *δ* = 0.07 nm, the radius of the probe molecule *r* was chosen to be equal to 0.11 nm (corresponding to the positronium radius).

In a study by Hofmann et al., free volume distributions obtained using computer simulations and PAL (positron annihilation lifetime) spectroscopy were compared [82]. It was established that the use of a test particle with a size of 0.11 nm in the simulations coincides with the experimental PAL spectroscopy data.

A probe molecule (test particle) was inserted at every point of the grid: if an overlap occurred between the test particle and any polymer atom, then the grid point was classified as “occupied”, otherwise it was classified as “free”. “Free” grid points which bordered each other were collected into groups. The volume of each cavity was calculated as the sum of the volumes of the test particles without their intersections with each other.

The separation performance of the polymer membrane could be characterized by permeability selectivity *α_P_,* diffusivity selectivity *α_D,_* and solubility selectivity *α_S_*. In the case of *α_S,_* it could be calculated as a ratio of the solubilities 
$S_{i}$
 and 
$S_{j}$
 of pure gases *i* and *j*, correspondingly:
(6)
$$\alpha_{S}=\frac{S_{i}}{S_{j}}$$


Characterization of the orientation of the PEI chains as a whole, relative to each other during the simulations, was carried out by calculating the order parameter *S_or_* for chain end-to-end vectors [83]. The value of *S_or_* was determined as the largest eigenvalue of the order tensor *S_αβ_*:
(7)
$$S_{\alpha \beta }=\frac{1}{N_{ch}}\sum_{i=1}^{N_{ch}}\left(\frac{3}{2}\vec{u_{i\alpha }}\;\vec{u_{i\beta }}-\frac{1}{2}\delta_{\alpha \beta }\right),$$

where *N_ch_* is the number of chains for which the calculation is performed, 
$\vec{u_{i}}$
 is the unit vector coinciding with the end-to-end vector of *i*-th chain, *δ* is the Kronecker delta, and *α*, *β* = *x*, *y*, or *z*.

## 3. Results and Discussion

### 3.1. Structure Ordering of BPDA-P3 Chains

The time dependence of the order parameter *S_or_* is shown in Figure 2.

During the first 16 μs of the simulations, the value of the parameter *S_or_* corresponding to the amorphous state (area 1 in Figure 2a) fluctuates strongly around 0.2 and does not exceed the value of 0.4. After 16 μs (up to 26 μs) of simulation, we observe the transitional state (area 2 in Figure 2a), where a gradual increase in the order parameter value occurs. After 26 μs (area 3 in Figure 2a) of the simulations, the polymer chains are ordered relative to each other, and order parameter *S_or_* reaches values close to 1 (0.97–0.98). Typical snapshots corresponding to the amorphous and ordered states of BPDA-P3 are shown in Figure 2b,c, respectively. The values of the order parameter *S_or,_* characterizing the amorphous and ordered state of the polymer, and the structural characteristics of the BPDA-P3 chains are presented in the Appendix A (see Appendix A, respectively). It can be seen from Appendix A that there is consistency in the values of order parameter *S_or_*, asphericity *b*, and the relative shape anisotropy *κ^2^* for BPDA-P3: in the amorphous state they are small, and in the ordered state they reach their limiting values with access to the plateau.

To select the time intervals separating the independent configurations of the system, we calculated the MSD of the chains’ centers of mass at 600 K in both the amorphous and ordered states (see Figure 3a). Analysis of the MSD curves allows us to determine one of the longest relaxation times characterizing the dynamic processes in the system, namely the displacement of the chains as a whole by their own sizes [45].

Analysis of the data presented in Figure 3a shows that the displacement of the chains as a whole by their own dimensions occurs at times of the order of 2 μs in an amorphous state. Meanwhile, in the ordered state, the chains diffuse by their own sizes at times of around 700 ns. These results suggest that the time interval of 2 μs between polymer configurations is sufficient to consider them as more or less independent configurations. Another interesting result is that the displacement of the chains as a whole for a fixed time in the ordered state is greater than that in the amorphous one, due to the “slipping” of the chains relative to each other in the ordered state. The slowdown of translational mobility in the amorphous state could be explained by the presence of entanglements of polymer chains with each other.

Having defined the time interval (2 μs) for obtaining independent configurations from the main 31 μs-long trajectory, we carried out a study of the free volume distributions in the amorphous and ordered states. The distributions of free volume elements were calculated for trajectories with a 5 ns duration at *T* = 600 K after 3, 5, and 7 μs of simulations (amorphous state) and 27, 29, and 31 μs of simulations (ordered state). The resulting distributions were averaged over each three systems (see Figure 3b). It can be seen from the dependencies obtained that ordering leads to free volume redistribution: the peak of the free volume (the most frequently appearing free volume cavity with a size *r*) shifts to the region of lower values—from cavity with *r* = 0.41 nm in amorphous state to *r* = 0.34 nm in ordered one. Furthermore, where values of *r* are greater than 0.5, the free volume distribution curve in the ordered state lies below the curve corresponding to the amorphous state. Temperature dependence of free volume distribution in amorphous and ordered states could be found in Appendix A.

The time dependence of the free volume fraction *Φ*, the average volume of cavity *V* in the simulation box, and the number of cavities of free volume *N* in the system were calculated in order to investigate the free volume redistribution during the ordering (see Figure 4a–c).

To obtain dependencies, showed on Figure 4, the entire trajectory was divided into fragments with a duration of 25 ns, and the values of *Φ*, *V* and *N* were calculated. After that, averaging was carried out over the 100 points obtained. After the first 2 μs of the simulation up to 16 μs, the values of *Φ*, *V* and *N* remain almost constant, weakly fluctuating near certain values, which corresponds to the amorphous state. However, after 16 μs, a gradual decrease in *Φ* and *V* in the system is observed, which is consistent with the increase in the order parameter (see Figure 2). The value of *N* in the system slightly increases (from ~684 to 712) during the ordering process. The mean values *N* were averaged over all points corresponding to the amorphous or ordered state respectively. Thus, as a result of structural ordering, the cavities gradually decrease in size, whereas their number becomes larger. Snapshots of instant configurations in the amorphous (after 3 μs) and ordered (after 29 μs) states are presented to visualize the changes in the free volume distributions (see Figure 5).

The snapshots confirm that, during the transition from an amorphous state to an ordered one, the pores become smaller in size, while their number increases.

As was mentioned above, PAL spectroscopy experiments [84,85] allow one to receive quantitative information about the size and distribution of the cavities of free volume in polymer sample. We can compare qualitatively our results with the available data for other PEIs. For example, free volume measurements for other PEIs were performed at room temperature (T = 298 K) in a recent study by Prof. Dingemans group [85]. For the series of PEIs, the average cavity volume was estimated between 0.001 till 0.029 nm^3^ [85] (compared with our results on Figure 4b, where the average volume of cavities changes from 0.1 nm^3^ till 0.06 nm^3^ during observing ordering at 600K). Taking into account significant difference in temperatures we can conclude the presence of qualitative agreement between the results obtained and known experimental data for other PEIs.

### 3.2. Transport Properties

The solubility values across a wide temperature range in amorphous and ordered states for CH_4_ and CO_2_ gases are presented in Figure 6a. To determine the solubility of the gases, as described in the Section 2.3, 10 ns-long additional computer simulations were carried out at each temperature studied.

It was found that, with a decrease in temperature, the solubility of both gases increases. Additionally, the solubility of both the gases considered is less in an ordered state than in an amorphous one. This decrease in solubility values could be explained by the redistribution of free volume in the bulk upon transition from an amorphous to an ordered state. The temperature dependence of solubility selectivity *α_S_* was calculated in both states (see Figure 6b). It was found that *α_S_* is greater in the state with a smaller fraction of free volume (ordered) than in the amorphous state. These results agree with those by the Robeson et al. [86], where it was shown that the solubility sites in the polymer become less accessible as the free volume decreases.

The diffusion coefficients for the gases at temperatures below Tg (≈487 K) in the range of 430 up to 480 K were calculated using the corresponding MSD curves (see Appendix A). Below 430 K, it is necessary to significantly increase the duration of the simulations in order to achieve a normal diffusion regime. The resulting temperature dependence of the diffusion coefficients D for both gases corresponds to the Arrhenius law (Equation (4)) (see Figure 7a,b). The latter fact indicates that the diffusion activation energy at the temperatures considered is preserved, and hence, the diffusion mechanism does not change. The diffusion activation energies for the CH_4_ and CO_2_ gases in amorphous and ordered states are presented in Table 1.

Comparing the resulting activation energies, we can conclude that in the ordered state, the diffusion activation energy is only slightly higher (or remains unchanged taking into account the calculation error) than in the amorphous one. In addition, *E_a_* for CO_2_ diffusion is lower in both states than *E_a_* for CH_4_, which is consistent with the values of the diffusion coefficient for these gases.

It was found that the diffusion coefficient for CO_2_ turns out to be higher than for CH_4_ which is more likely due to the difference in the effective diameter of the molecules. This effect is preserved for both amorphous and ordered states (see Figure 7a,b).

Comparison of the diffusion coefficient values in the amorphous and ordered states leads to the conclusion that *D* values in ordered and in amorphous states are very similar to one another (see Figure 7c). It can be assumed that the diffusion of gas molecules in the ordered state in the direction of the ordering of the polymer chains could differ from their diffusion in other directions. Since the direction of the ordering of polymer chains was not associated with the axes of the simulation box, but appeared randomly, we constructed a new orthogonal basis, the first axis of which corresponds to the direction of ordering, while the other two are orthogonal to it. The results of studying the anisotropy of the gas diffusion are presented in Figure 8. The direction of the polymer chain ordering (the director of ordering) was calculated in the oriented state (after 30 μs of simulation) as an eigenvector corresponding to the largest eigenvalue of the order tensor *S_αβ_* (see Equation (7)). It was found that this direction remains practically unchanged during the simulation time after the onset of ordering.

Analyzing Figure 8, one can see an increase in diffusion along the direction of ordering (x_new_) compared to the other two directions (y_new_ and z_new_). As such, the diffusion of gases occurs predominantly along the direction of the ordering of polymer chains, while on average there is practically no difference in the diffusion coefficient values in the amorphous and ordered states (see Figure 7c). Similar graphs were obtained at all other temperatures ranging between 430 K and 480 K (not shown). At the same time, it should be noted that neither conducting nor nonconducting domains appear in the system, as, for example, in the K.-Kh. Shen et al. study that showed an overall reduction in selective penetrant diffusion in systems with a cylindrical morphology to 1/3 compared to a bulk system [87]. In this regard, due to the weak influence of structural ordering on the diffusion coefficient of the gases in the system, the permeability will be more influenced by the gas solubility than by the diffusion coefficient. Since solubility decreases during structural ordering, it leads to a decrease in permeability (see Figure 9).

## 4. Conclusions

Based on the results obtained, we can conclude that a change in the supramolecular structure of the polymer sample, that is to say, in the structural ordering of polymer chains of semi-crystalline PEI BPDA-P3 in bulk, leads to a decrease in free volume and its redistribution with an increasing fraction of smaller free volume cavities. Such redistribution leads to a change in the solubility of the gases: in the process of structural ordering, the solubility of the gases decreases. However, structural ordering does not affect the average diffusion coefficient of the gases. The formation of smaller cavities does not in practice restrict the motion of gases through the polymer, since the sizes of the considered gas molecules are comparable with the cavity sizes. Despite the fact that the movement of the gas molecules in an amorphous and ordered system averaged over all three coordinate axes is practically the same, the local diffusion of gas in the direction of the ordering of polymer chains is significantly higher than the diffusion in the normal direction. Combining these factors (since permeability is the result of the multiplication of solubility and the diffusion coefficient), the permeability of BPDA-P3 decreases as the structure ordering increases.

Therefore, by changing the supramolecular structure of polymer chains, it is possible to change the transport properties of the systems under consideration. The «supramolecular structure»—«properties» relationship thus established could later be used in the development of new PEI-based membrane materials with controlled properties.

## Figures and Tables

**Figure 1 membranes-12-00856-f001:**
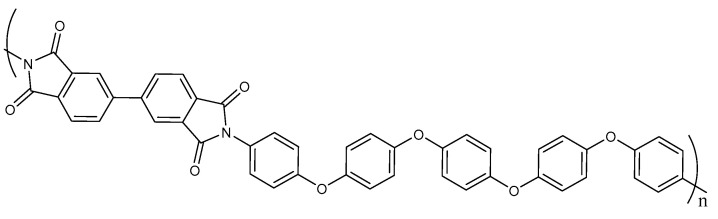
The chemical structure of the repeating unit of the thermoplastic PEI BPDA-P3 [30].

**Figure 2 membranes-12-00856-f002:**
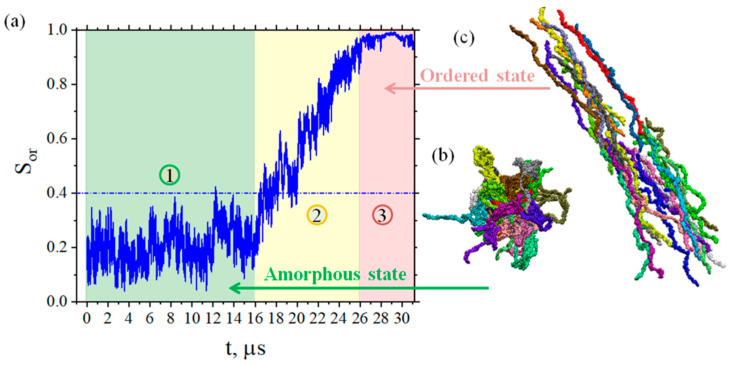
(**a**) Time dependence of the order parameter *S_or_* for the end-to-end vector of BPDA-P3 chains at 600 K. Area 1 denotes the amorphous state of the polymer (green color), area 2 denotes the transitional state (yellow color), and area 3 denotes the ordered state (red color). (**b**) Snapshots of a typical instantaneous configuration of the system in an amorphous state (after 5 ns of simulations). (**c**) Snapshots of a typical instantaneous configuration of the system in an ordered state (after 31 μs of simulations). Each chain has its own color.

**Figure 3 membranes-12-00856-f003:**
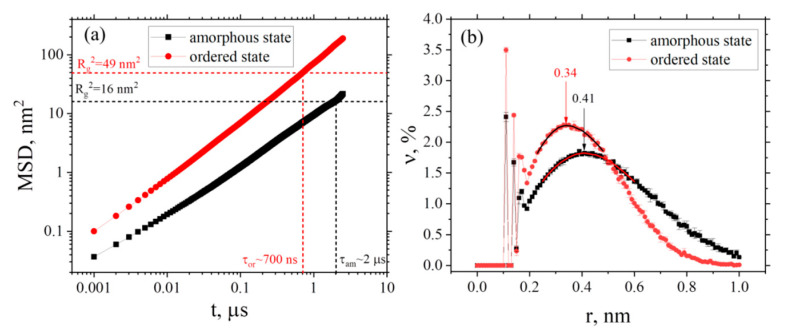
(**a**) Time dependence of the MSD of BPDA-P3 polymer chains’ centers of mass in amorphous (black line and filled squares) and ordered (red line and filled circles) states. The dotted line indicates the radius of gyration for the polymer chains in amorphous (black) and ordered (red) states, and the times at which the polymer chains shift to their own dimensions. (**b**) Free volume distribution (% of the total fraction of free volume) in amorphous (black line and filled squares) and ordered (red line and filled circles) states.

**Figure 4 membranes-12-00856-f004:**
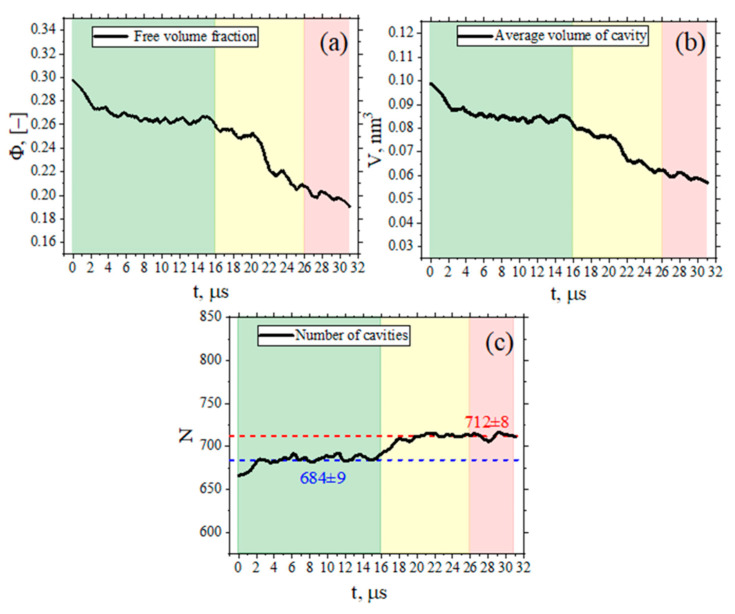
Time dependence of (**a**) the free volume fraction *Φ*, (**b**) the average volume of cavity *V* and (**c**) the number of cavities in the simulation box *N*. Dashed lines denote the average number of cavities in the amorphous (blue) and ordered (red) states with the mean square error.

**Figure 5 membranes-12-00856-f005:**
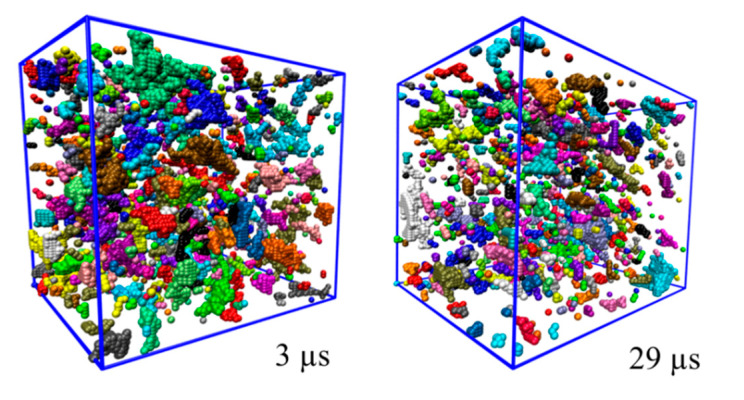
Snapshots of typical instant configurations of free volume distribution in the simulation box after 3 (**left**) and 29 (**right**) µs respectively. Each pore has its own color.

**Figure 6 membranes-12-00856-f006:**
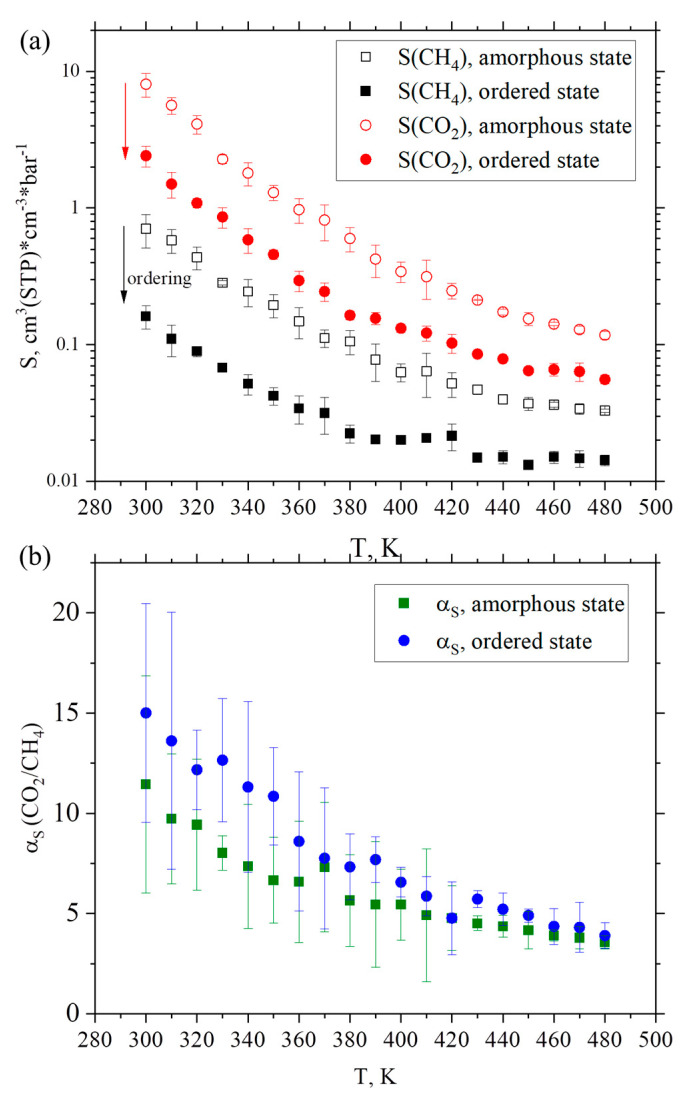
(**a**) Temperature dependence of solubility *S* for CH_4_ (black squares) and CO_2_ (red circles) gases in amorphous (open symbols) and ordered (filled symbols) states. (**b**) Temperature dependence of solubility selectivity *α_S_* for CH_4_/CO_2_ pair in amorphous and ordered states.

**Figure 7 membranes-12-00856-f007:**
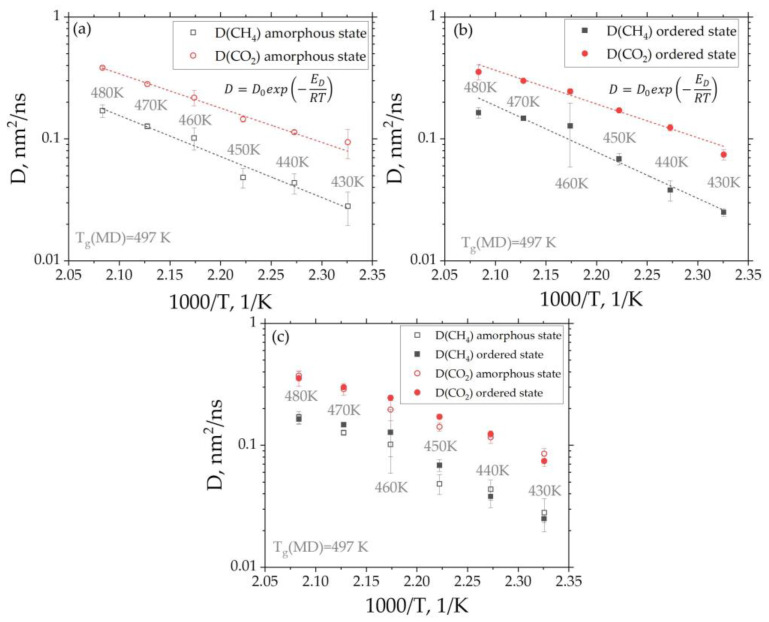
Dependence of the diffusion coefficients of the CH_4_ and CO_2_ gases in amorphous (**a**) and ordered (**b**) states in BPDA-P3 on the reciprocal temperature. Dashed fitting lines were obtained by approximation of the diffusion coefficient values using the Arrhenius law (Equation (4)). (**c**) Dependence of the diffusion coefficients on the reciprocal temperature for both gases in amorphous and ordered states. The BPDA-P3 glass transition temperature shown in (**a**–**c**) was obtained in our previous study using the all-atom Molecular Dynamics simulations.

**Figure 8 membranes-12-00856-f008:**
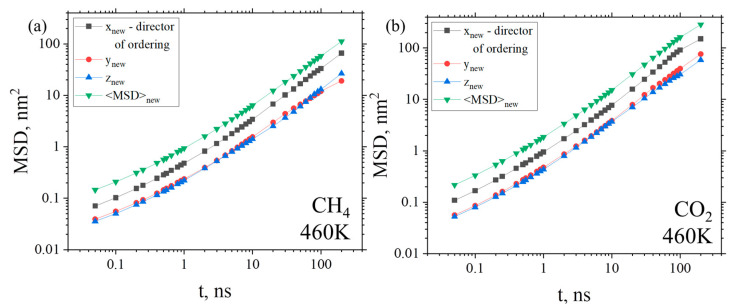
Time dependence of MSD for (**a**) CH_4_ and (**b**) CO_2_ gases in an ordered state for BPDA-P3 along the “new” axes: the one axis corresponds to the direction of ordering (black curve) and the two other axes are perpendicular to the first (red and blue curve). The green curve denotes the total MSD as the sum of MSD along three components (x_new_, y_new_ and z_new_) in the “new” coordinate system. The graphs presented correspond to the system obtained after 29 μs of simulations (ordered state) at cooling to T = 460 K.

**Figure 9 membranes-12-00856-f009:**
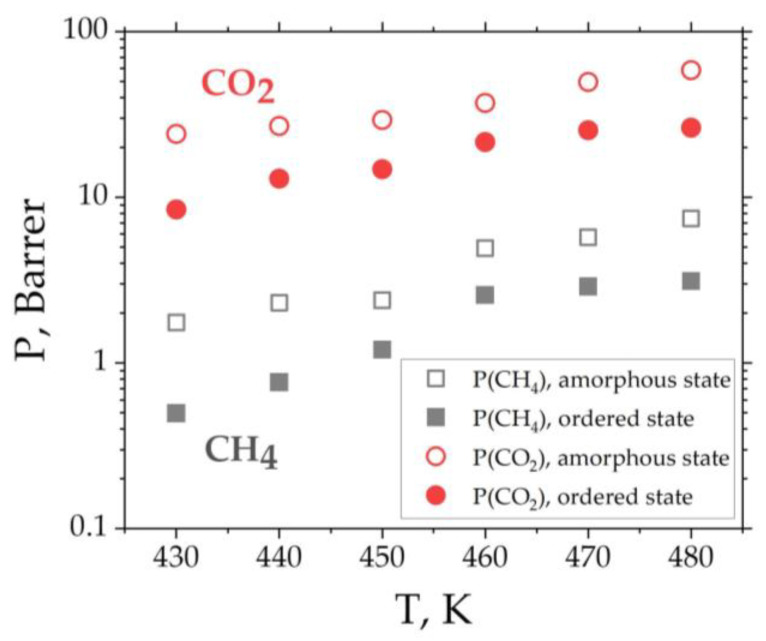
Temperature dependence of CH_4_ and CO_2_ permeability *P* of BPDA-P3 in amorphous and ordered states.

**Table 1 membranes-12-00856-t001:** Activation energy *E_a_* of diffusion of the CH_4_ and CO_2_ gases for BPDA-P3 in amorphous and ordered states.

	Amorphous State		Ordered State	
	CH_4_	CO_2_	CH_4_	CO_2_
*E_a_*, kJ/mol	64.4 ± 5.4	53.8 ± 1.1	72.6 ± 4.2	53.0 ± 3.6

## Data Availability

Not applicable.

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
