# Peer review of "The Transport Properties of Semi-Crystalline Polyetherimide BPDA-P3 in Amorphous and Ordered States: Computer Simulations"

_membranes, 2022, doi:10.3390/membranes12090856_

Round 1
Reviewer 1 Report
In the current study, the effect of polymer chain ordering on the transport properties of the polymer membrane was examined for the semi-crystalline (PEI) BPDA-P3 by molecular dynamics simulations. After accounting the solubility and diffusivity of gas molecules in amorphous and ordered morphologies, the decrease in permeability of BPDA-P3 in ordered state can be attributed to the decreased solubility due to decreasing of free volume and its redistribution with increasing fraction of smaller free volume cavities. The paper is well-written, the logic is self-explanatory, I am really enjoying reading this manuscript. However, before the study can be accepted by Membranes, I have the following concerns that need to be addressed by the authors.
1. According to the authors, the 31 μs simulation cost one and a half years, however, the disordered to ordered state transition is out of the scope of the current study, is it really necessary to conduct simulation in such a way? If I understand correctly, the authors only want to obtain uncorrelated configurations for the amorphous and ordered states, there are multiple ways to obtain desired configurations, such multiscale simulation methodology, meaning construct the desired configuration at coarse-grained scale, and the back-map to the atomistic scale. For examples, from recent works by Venkat Ganesan from UT Austin, they have developed a multiscale simulation framework [ACS Macro Letters 8 (9), 1096-1101; Macromolecules 54 (11), 4997-5010], combining both coarse-grained and atomistic simulations to study the influence of morphology on the ion transport properties of polyelectrolytes.
2. It is extremely rare to see atomistic simulation without partial charges. I understand that such implementation could reduce the simulation time dramatically. However, after obtaining the configurations for amorphous and ordered states, shouldn’t the authors re-introduce the partial charges for calculating the solubility and diffusivity?
3. For the calculation of solubility, the authors use the changes in potential energy to track excess chemical potential µex, could the authors provide more details of calculating the Delta U? The raw data should be provided in SI. Moreover, to calculate Delta U, the authors have employed a system with insertion of 10 gas molecules, is this the most typical/practical way used in similar works? As has been mentioned by the authors: “the solubility S may be estimated using Widom’s test-particle insertion method”, however, from what I know, the GCMC method or free energy method are the generally used methods. Could the authors comment on this?
4. If I understand correctly, the values for equation 7 are ranging from -0.5 to 1.0, however, the results turn on to be varying from 0.0 to 1.0, could the authors comment on this?
5. The position of the caption of Figure 5 is incorrect, should appear directly after the Figure.
6. For the diffusion coefficient, could the author provide more details, such as the fitting range and the superscript (beta) of t (D~tbeta) in the log-log plot?
7. For the diffusion coefficient, I think the authors shouldn’t compare the overall values directly between amorphous and ordered states, since the three eigenvalues of D are different in ordered state, I would like to see that the authors directly compare the D after accounting for the dimensionality effects, as has been suggested by Lisa M. Hall and many others’ works [ACS Macro Lett. 2018, 7, 1092−1098]. In detail, from the presented results, I believe that the D in ordered state is one dimensional, so the true D for ordered state should be the maximum eigenvalue of D multiple a factor of 3. Could the authors comment on this?

Author Response
Reviewer 1
We would like to thank the reviewer for high estimation of our study and valuable comments, all of which were taken into account in the revised version of the manuscript. We provide the detailed answers below.
- According to the authors, the 31 μs simulation cost one and a half years, however, the disordered to ordered state transition is out of the scope of the current study, is it really necessary to conduct simulation in such a way? If I understand correctly, the authors only want to obtain uncorrelated configurations for the amorphous and ordered states, there are multiple ways to obtain desired configurations, such multiscale simulation methodology, meaning construct the desired configuration at coarse-grained scale, and the back-map to the atomistic scale. For examples, from recent works by Venkat Ganesan from UT Austin, they have developed a multiscale simulation framework [ACS Macro Letters 8 (9), 1096-1101; Macromolecules 54 (11), 4997-5010], combining both coarse-grained and atomistic simulations to study the influence of morphology on the ion transport properties of polyelectrolytes.
The main reason for rather long MD simulations was related to our interest in the dynamics of free volume redistribution. As we know from our previous study [1], the process of PEI chain ordering can be divided into two stages: at the first stage, only small fragments of chains begin to orient, and then more and more fragments are ordered (after 20 μs of simulations in the current study), which leads to the orientation of polymer chains as a whole. We found that a significant redistribution of the free volume is observed only at the second stage (see Figures 3 and 4 for details).
The use of coarse-grained (CG) simulations, indeed, can significantly reduce the time required for computer simulations compared to all-atom simulations. This approach is often used in the study of various polymers, in particular polyetherimides (PEI), and nanocomposites based on them [2–6]. One of the ways to move from a CG model to an all-atom one that consider the details of the chemical structure of the polymer is reverse mapping (RM) procedure [7]. For example, in the studies by Prof. Ganesan with colleagues mentioned in the review, the multiscale simulations have been performed. These simulations consisted of three successive stages: «ideal» CG bead-spring model simulations, with subsequent transition to «real» model of higher chemical fidelity and RM procedure where the details of chemical structure were introduced to CG morphology [8,9]. This approach has been used to study the ion transport in block copolymeric ionic liquids. Interesting results were obtained when comparing density profiles of coarse-grained and atomistic structures before and after RM. The use of RM for epoxy networks [10], atactic polystyrene [11], polyethylene terephthalate [7] and other polymers was also proposed. However, the chemical structure of the mentioned systems is much simpler than that of heterocyclic PEI BPDA-P3. To the best of our knowledge, there is only a small number of studies where CG simulations followed by RM are used to investigate PEI [5,12,13].
It should be noted that the use of CG models in computer simulations (before transition to all-atom models using the RM procedure) can lead to the loss of the detailed (atomistic) description of molecules, which is available only in all-atom simulations, and may be of decisive importance in studying the transport properties of polymeric membranes. It is also not obvious that the ordering that might be observed during the CG simulations and subsequent RM procedure would exactly match the ordering obtained in the atomistic simulations. Since, as shown in our work, the effects associated with a change in the free volume occur on scales of the order of nanometers and even insignificant changes associated with the inaccuracy of reproducing the ordered structure can lead to incorrect calculations of the transport properties.
Moreover, to perform CG simulations, it is necessary to create and validate a suitable CG model of the considered PEI. However, it requires a long all-atom trajectory which was performed in this study. The use of CG model is a non-trivial task and is certainly an important independent area of our further research. Such a study is also planned in our group, since we have successful experience in the development and validation of CG PEI models [6].
Thus, in our research, the creation of a new CG model with the development of the RM method, most likely, would not give a gain in time and labor costs. At the same time, it is worth noting that reviewer' proposal to use multiscale simulations with RM approach is certainly may be very powerful and will be an important continuation of our work in the future.
We have made changes following the reviewer’s suggestion on the pages 4 and 5.
- It is extremely rare to see atomistic simulation without partial charges. I understand that such implementation could reduce the simulation time dramatically. However, after obtaining the configurations for amorphous and ordered states, shouldn’t the authors re-introduce the partial charges for calculating the solubility and diffusivity?
Definitely, accounting of partial charges leads to a significant increase in simulation time. However, as was shown earlier in our works, qualitative agreement between simulational and experimental data on structural and mechanical properties can also be observed when using models that do not take into account the presence of partial charges [14,15]. Thus, in the absence of strongly polar groups in the polymer, the contribution of partial charges may not be very important. Moreover, our investigation of the PEI transport properties without taking into account partial charges provide a quantitative agreement between the transport properties between the data obtained by simulation and in the experiment [16]. In addition, if gas molecules do not have strong charges, then the absence of partial charges in polymers can be justified, due to the absence of strong electrostatic "gas" - "polymer" interactions.
Nevertheless, a study of the effect of partial charges on transport properties caused by the redistribution of free volume really needs to be done. Such problem is the subject of our separate study, the really intriguing results of which are now being prepared for publication as a full article with the preliminary title Influence of the cooling rate on gas solubility in a thermostable polyimide and its free volume: molecular dynamic simulation by M. V. Andreeva, I. V. Volgin, S. V. Larin, L. I. Klushin, and S. V. Lyulin.
The additional comment related to the question have been added on the page 5.
- For the calculation of solubility, the authors use the changes in potential energy to track excess chemical potential µex, could the authors provide more details of calculating the Delta U? The raw data should be provided in SI. Moreover, to calculate Delta U, the authors have employed a system with insertion of 10 gas molecules, is this the most typical/practical way used in similar works? As has been mentioned by the authors: “the solubility S may be estimated using Widom’s test-particle insertion method”, however, from what I know, the GCMC method or free energy method are the generally used methods. Could the authors comment on this?
In the manuscript probably we have omitted some details describing the calculation procedure of excess chemical potential µex. GROMACS software with tpi integrator [17] was used to perform test particle insertion method [18]. The output file contains only moving averaged values of µex over all considered trajectory calculated from ΔU values, corresponding to the insertion of the one test particle, takes place within the GROMACS itself. So, the final value of µex is used to calculate solubility S using equation 2 in the manuscript.
Indeed, the calculation of the solubility of gases can be implemented by several widely used methods, such as Grand Canonical Monte Carlo (GCMC), free energy method, and Widom (test particle) insertion method, each of which is used for specialised tasks.
Grand Canonical Monte Carlo (GCMC) is currently commonly-used method to calculate gas sorption [19,20]. Sorption mechanism based on GCMC and Metropolis method works relying on accepting or rejecting new molecules regarding their trial moves. Solubility coefficient depends on the interaction of penetrant molecules with polymer matrix as a function of a thermodynamic factor [21].
Free energy calculation is separated into three stages. First, series of intermediate states are generated, then each state is properly sampled with MC or MD simulations, and finally one of methods is used to analyze and estimate the free energy difference between the states of interest [22].
In Test Particle Insertion (TPI) approach (Widom`s method) a test molecule is inserted randomly into a polymer configuration, and the excess chemical potential is calculated directly. Such method is very similar to GCMC due to particle insertion randomness. To calculate the excess chemical potential, first molecular dynamics simulations are performed at a specified temperature and pressure of the polymer, without any gas molecules. After that several configurations are extracted from the dynamic simulation and used to insert the test particles. A test molecule is repeatedly inserted in random positions and random orientations into the selected configurations, and the energy change due to insertions is calculated [23]. This approach allows one to determine the solubility of small molecules in different systems with good agreement with experimental results [16,19,24].
Corresponding explanations have been added to the revised version of the manuscript, please see page 7.
- If I understand correctly, the values for equation 7 are ranging from -0.5 to 1.0, however, the results turn on to be varying from 0.0 to 1.0, could the authors comment on this?
In our study, to calculate the degree of ordering of BPDA-P3 polymer chains, we used the order parameter Sor, which is calculated through the tensor Sαβ (Equation 7 in the manuscript) regardless of the chosen axis. In fact, it was close in meaning to the nematic ordering parameter. This order parameter can take values from 0 to 1, unlike the order parameter S(r), which is calculated through the Legendre polynomial as 3/2<cos2θ(r)>-1/2 , where cos2θ(r) is the mean square of the angle cosine between vector associated with polymer chains and an axis along which ordering is considered. In the latter case, indeed, S(r) can take values from -0.5 to 1.
As noted in SI, the approach to calculating the order parameter Sor proposed in the manuscript leads to Sor values from 0 to 1. This is due to the fact that in a completely amorphous system, the vectors for which this parameter is calculated turn out to be distributed in a sphere in all directions (see Figure S1 in Supplementary Materials). Thus, there will always be a vector opposite to the given one. In this regard, the set of such opposite vectors leads to the minimum possible value Sor=0. Moreover, in the proposed version of the calculation of the order parameter, the value of Sor corresponds to the largest eigenvalue of the tensor ??? (Equation 7 in the manuscript), thus, the largest value of the order parameter cannot be negative [25].
The comment related to this question have been added on the page S1 in Supplementary Materials.
- The position of the caption of Figure 5 is incorrect, should appear directly after the Figure.
Thank you for the comment. Indeed, the position of Figure 5 is incorrect; we will definitely put it in the proper place in the manuscript (page 12).
- For the diffusion coefficient, could the author provide more details, such as the fitting range and the superscript (beta) of t(D~tbeta) in the log-log plot?
The diffusion coefficients in the amorphous and ordered states were calculated from the mean-square displacement (MSD) curves for CO2 и CH4 gas molecules, see Figure S4 and Figure S5 in Supplementary Materials. MSD of gas molecules was calculated from trajectories with 500 ns duration. The initial configurations for the trajectories were created by BPDA-P3 cooling from 600 K to the considered temperatures in the range from 480 to 430 K after 3, 5, 7, 27, 29 and 31 µs of simulations. The presented graphs show the MSD curves along the trajectory from 0 to 250 ns and the dotted line indicates guides to the eye which represent the linear MSD dependence in the normal diffusive regime (slope=1). On the MSD curve in log-log coordinates, an area with slope=1 was selected. This section of curve was approximated by a linear function y=kt+b. The resulting b value was used to calculate the diffusion coefficient.
Since in logarithmic coordinates the equation <Δr2>=6Dt can be rewritten as: log10<Δr2>=log10t+log106D, addend log106D will correspond to the value of the parameter b in the linear approximation. Then we get that .
Relevant explanations have been added to Supplementary Materials, please see page S5 .
- For the diffusion coefficient, I think the authors shouldn’t compare the overall values directly between amorphous and ordered states, since the three eigenvalues of D are different in ordered state, I would like to see that the authors directly compare the D after accounting for the dimensionality effects, as has been suggested by Lisa M. Hall and many others’ works [ACS Macro Lett. 2018, 7, 1092−1098]. In detail, from the presented results, I believe that the D in ordered state is one dimensional, so the true D for ordered state should be the maximum eigenvalue of D multiple a factor of 3. Could the authors comment on this?
The main difference between our study and the study by Lisa M. Hall is the difference in considered types of penetrants. In the proposed study simulations with addition of selective penetrants (which preferentially interact with one microphase) are performed while in our study the penetrant is not selective [26]. Moreover, the authors consider diffusion through the structures with different morphologies (cylinders, lamellae, and gyroids) containing two types of domains – conducting and nonconducting. Thus, in the systems with a cylinder type of morphology, the diffusion of the penetrant will be strongly limited and will occur in one direction. This explains the overall diffusion reduction to 1/3 of the diffusion in a bulk system.
In our study, diffusion of nonselective penetrants in ordered BPDA-P3 system occurs in all three directions, see Figure 8 in the manuscript. At the same time, we note that diffusion in the direction of ordering turns out to be much larger compared to diffusion in the other two directions.
The comment related to this question have been added on the page 15.
We have made appropriate changes in the manuscript according to all the reviewer’s suggestions.
References:
- Larin, S. V.; Nazarychev, V.M.; Dobrovskiy, A.Y.; Lyulin, A. V.; Lyulin, S. V. Structural ordering in SWCNT-polyimide nanocomposites and its influence on their mechanical properties. Polymers. 2018, 10, 1245, doi:10.3390/polym10111245.
- Kumar, A.; Sudarkodi, V.; Parandekar, P. V; Sinha, N.K.; Prakash, O.; Nair, N.N.; Basu, S. Adhesion between a rutile surface and a polyimide: a coarse grained molecular dynamics study. Model. Simul. Mater. Sci. Eng. 2018, 26, 035012, doi:10.1088/1361-651X/aaa9e2.
- Pandiyan, S.; Parandekar, P. V.; Prakash, O.; Tsotsis, T.K.; Basu, S. Systematic Coarse Graining of a High-Performance Polyimide. Macromol. Theory Simulations 2015, 24, 513–520, doi:10.1002/mats.201500009.
- Wen, C.; Odle, R.; Cheng, S. Coarse-Grained Molecular Dynamics Modeling of a Branched Polyetherimide. Macromolecules 2021, 54, 143–160, doi:10.1021/acs.macromol.0c01440.
- Odegard, G.M.; Clancy, T.C.; Gates, T.S. Modeling of the mechanical properties of nanoparticle/polymer composites. Polymer. 2005, 46, 553–562, doi:10.1016/j.polymer.2004.11.022.
- Volgin, I.V.; Larin, S.V.; Lyulin, A.V.; Lyulin, S.V. Coarse-grained molecular-dynamics simulations of nanoparticle diffusion in polymer nanocomposites. Polymer. 2018, 145, 80–87, doi:10.1016/j.polymer.2018.04.058.
- Krajniak, J.; Zhang, Z.; Pandiyan, S.; Nies, E.; Samaey, G. Reverse mapping method for complex polymer systems. J. Comput. Chem. 2018, 39, 648–664, doi:10.1002/jcc.25129.
- Zhang, Z.; Krajniak, J.; Keith, J.R.; Ganesan, V. Mechanisms of Ion Transport in Block Copolymeric Polymerized Ionic Liquids. ACS Macro Lett. 2019, 8, 1096–1101, doi:10.1021/acsmacrolett.9b00478.
- Zhang, Z.; Krajniak, J.; Ganesan, V. A Multiscale Simulation Study of Influence of Morphology on Ion Transport in Block Copolymeric Ionic Liquids. Macromolecules 2021, 54, 4997–5010, doi:10.1021/acs.macromol.1c00025.
- Gavrilov, A.A.; Komarov, P. V.; Khalatur, P.G. Thermal properties and topology of epoxy networks: A multiscale simulation methodology. Macromolecules 2015, 48, 206–212, doi:10.1021/ma502220k.
- Spyriouni, T.; Tzoumanekas, C.; Theodorou, D.; Müller-Plathe, F.; Milano, G. Coarse-grained and reverse-mapped united-atom simulations of long-chain atactic polystyrene melts: Structure, thermodynamic properties, chain conformation, and entanglements. Macromolecules 2007, 40, 3876–3885, doi:10.1021/ma0700983.
- Clancy, T.C.; Hinkley, J.A. Coarse-Grained and Atomistic Modeling of Polyimides. NASA Tech. Reports Serv. 2004.
- Clancy, T.C. Multi-scale modeling of polyimides. Polymer. 2004, 45, 7001–7010, doi:10.1016/j.polymer.2004.08.009.
- Larin, S. V.; Falkovich, S.G.; Nazarychev, V.M.; Gurtovenko, A.A.; Lyulin, A. V.; Lyulin, S. V. Molecular-dynamics simulation of polyimide matrix pre-crystallization near the surface of a single-walled carbon nanotube. RSC Adv. 2014, 4, 830–844, doi:10.1039/C3RA45010D.
- Nazarychev, V.M.; Dobrovskiy, A.Y.; Larin, S. V.; Lyulin, A. V.; Lyulin, S. V. Simulating local mobility and mechanical properties of thermostable polyimides with different dianhydride fragments. J. Polym. Sci. Part B Polym. Phys. 2018, 56, 375–382, doi:10.1002/polb.24550.
- Volgin, I. V.; Andreeva, M. V.; Larin, S. V.; Didenko, A.L.; Vaganov, G. V.; Borisov, I.L.; Volkov, A. V.; Klushin, L.I.; Lyulin, S. V. Transport properties of thermoplastic R-BAPB polyimide: Molecular dynamics simulations and experiment. Polymers (Basel). 2019, 11, 1775, doi:10.3390/polym11111775.
- Abraham, M.J.; van der Spoel, D.; Lindahl, E.; Hess, B. GROMACS User Manual version 5.0.4. In; Berlin, 2015; p. 312 ISBN 3032783534.
- Web Page: Test particle insertion Available online: https://gaseri.org/en/tutorials/gromacs/6-tpi/.
- Neyertz, S.; Brown, D. Single- and mixed-gas sorption in large-scale molecular models of glassy bulk polymers. Competitive sorption of a binary CH4/N2 and a ternary CH4/N2/CO2 mixture in a polyimide membrane. J. Memb. Sci. 2020, 614, 118478, doi:10.1016/j.memsci.2020.118478.
- Neyertz, S.; Brown, D.; Salimi, S.; Radmanesh, F.; Benes, N.E. Molecular Characterization of Membrane Gas Separation under Very High Temperatures and Pressure: Single- and Mixed-Gas CO2/CH4 and CO2/N2 Permselectivities in Hybrid Networks. Membranes. 2022, 12, 526, doi:10.3390/membranes12050526.
- Riasat Harami, H.; Riazi Fini, F.; Rezakazemi, M.; Shirazian, S. Sorption in mixed matrix membranes: Experimental and molecular dynamic simulation and Grand Canonical Monte Carlo method. J. Mol. Liq. 2019, 282, 566–576, doi:10.1016/j.molliq.2019.03.047.
- Liu, H.; Dai, S.; Jiang, D.E. Solubility of gases in a common ionic liquid from molecular dynamics based free energy calculations. J. Phys. Chem. B 2014, 118, 2719–2725, doi:10.1021/jp500137u.
- Eslami, H.; Müller-Plathe, F. Molecular dynamics simulation of sorption of gases in polystyrene. Macromolecules 2007, 40, 6413–6421, doi:10.1021/ma070697+.
- Hossain, S.; Kabedev, A.; Parrow, A.; Bergström, C.A.S.; Larsson, P. Molecular simulation as a computational pharmaceutics tool to predict drug solubility, solubilization processes and partitioning. Eur. J. Pharm. Biopharm. 2019, 137, 46–55, doi:10.1016/j.ejpb.2019.02.007.
- Anwar, M.; Turci, F.; Schilling, T. Crystallization mechanism in melts of short n-alkane chains. J. Chem. Phys. 2013, 139, 214904–214908, doi:10.1063/1.4835015.
- Shen, K.H.; Brown, J.R.; Hall, L.M. Diffusion in Lamellae, Cylinders, and Double Gyroid Block Copolymer Nanostructures. ACS Macro Lett. 2018, 7, 1092–1098, doi:10.1021/acsmacrolett.8b00506.

Reviewer 2 Report
The paper presents an original and interesting study on the transport properties of amorphous and ordered regions of heterocyclic Poly(ether imide) compounds.
The manuscript is well written and although the findings are not of outstanding relevance, it will contribute to shed some light on the gas permeability mechanism of PEI.
The results are presented clearly and lead to some reasonable conclusions which are well reported in the final section of the manuscript.
I have one comment for the authors:
It would be interesting to see how your results for fractional free volume predictions compare with experimental data, for example by means of the standard technique used - PAL (Positron annihilation lifetime) spectroscopy.
Also, it would be interesting to carry out simulations predicting fractional free volume in a range of temperature more representative of the service life of a PEI membrane. Do you have any data at all on this?
For further reference :
Free Volume in Poly(ether imide) Membranes Measured by Positron Annihilation Lifetime Spectroscopy and Doppler Broadening of Annihilation Radiation
Zeljka P. Madzarevic, Henk Schut, Jakub ÄŒížek, and Theo J. Dingemans
Macromolecules 2018 51 (23), 9925-9932
DOI: 10.1021/acs.macromol.8b01723
R Ramani and S Alam 2015 J. Phys.: Conf. Ser. 618 012035
Author Response
Reviewer 2
First of all, we would like to thank the reviewer for valuable comments, questions and appreciation of our work. Detailed answers which were taken into account in the revised version of manuscript are presented below.
- It would be interesting to see how your results for fractional free volume predictions compare with experimental data, for example by means of the standard technique used - PAL (Positron annihilation lifetime) spectroscopy.
For further reference : Free Volume in Poly(ether imide) Membranes Measured by Positron Annihilation Lifetime Spectroscopy and Doppler Broadening of Annihilation Radiation. Zeljka P. Madzarevic, Henk Schut, Jakub ÄŒížek, and Theo J. Dingemans. Macromolecules 2018 51 (23), 9925-9932. DOI: 10.1021/acs.macromol.8b01723; R Ramani and S Alam 2015 J. Phys.: Conf. Ser. 618 012035
We would like to thank reviewer for this very important comment. Indeed, PAL (Positron annihilation lifetime) spectroscopy experiments allow one to receive quantitative information about the size and distribution of the voids of free volume in polymer sample. However, to the best of our knowledge, experimental data of BPDA-P3 transport properties are currently not published in the literature that's why it is difficult to compare the obtained data of computer simulations with experimental data directly. However, we can compare qualitatively our results with the results for other PEIs. For example, free volume measurements for other PEIs have been performed by Prof. Ramani [1] and at room temperature (T=298 K) in recent study by Prof. Dingemans group [2], kindly provided by reviewer. For the series of PEIs, the Average Volume of Free-Volume elements were estimated between 0.001 till 0.029 nm3 [2] (please, compare with our results on Figure 4b, where average volume of cavities change from 0.1 nm3 till 0.06 nm3 during observing ordering at 600K). Taking into account significant difference in temperatures we can conclude the presence of agreement between obtained results and known experimental data for other PEIs.
Moreover, for free volume calculations with test particle insertion method we used the value of the test particle radius r = 1.1 Å, as the recommended by Prof. Hoffman et al. [3], which corresponds to the orthopositronium (o-Ps) radius in the study of the free volume by the positron annihilation method. In another study of this group, free volume distributions obtained from computer simulations and PAL spectroscopy were compared [4]. It has been established that the use of a test particle with a size of 1.1 Å (the size of a positronium molecule) in the simulations gives a good agreement with the experimental PAL spectroscopy data.
In the future, we plan to test the obtained results and conclusions experimentally.
The additional comment related to the question have been added on the pages 8 and 12.
- Also, it would be interesting to carry out simulations predicting fractional free volume in a range of temperature more representative of the service life of a PEI membrane. Do you have any data at all on this?
Owing to the important suggestion of the reviewer, we have carried out additional calculations of free volume distributions. After cooling from 600 to 300 K of amorphous and ordered systems (after 5 and 27 µs of simulation, respectively), free volume distribution (% of the total fraction of free volume) was calculated along a 5ns-long trajectory similar to the distributions in Figure 3b in the manuscript. Corresponded graphs are given in the Supplementary Materials, see Figure S3. It can be concluded from the graphs that with decreasing temperature, the distributions shift to the region of smaller values of free volume element`s sizes, both in the amorphous and in the ordered state.
The comment related to this question have been added on the page 10 in the manuscript and on the page S4 in Supplementary Materials.
We have made appropriate changes in the manuscript according to all the reviewer’s suggestions.
References:
- Ramani, R.; Alam, S. Free volume study on the miscibility of PEEK/PEI blend using positron annihilation and dynamic mechanical thermal analysis. J. Phys. Conf. Ser. 2015, 618, 1–5, doi:10.1088/1742-6596/618/1/012035.
- Madzarevic, Z.P.; Schut, H.; ÄŒížek, J.; Dingemans, T.J. Free Volume in Poly(ether imide) Membranes Measured by Positron Annihilation Lifetime Spectroscopy and Doppler Broadening of Annihilation Radiation. Macromolecules 2018, 51, 9925–9932, doi:10.1021/acs.macromol.8b01723.
- Hofmann, D.; Heuchel, M.; Yampolskii, Y.; Khotimskii, V.; Shantarovich, V. Free volume distributions in ultrahigh and lower free volume polymers: Comparison between molecular modeling and positron lifetime studies. Macromolecules 2002, 35, 2129–2140, doi:10.1021/ma011360p.
- Hofmann, D.; Entrialgo-Castano, M.; Lerbret, A.; Heuchel, M.; Yampolskii, Y. Molecular modeling investigation of free volume distributions in stiff chain polymers with conventional and ultrahigh free volume: Comparison between molecular modeling and positron lifetime studies. Macromolecules 2003, 36, 8528–8538, doi:10.1021/ma034971l.

Round 2
Reviewer 1 Report
The authors have addresses all the questions I had before, I am agree for accepting the paper to be published on Membranes.